# Finite Element Analysis of Maxillary Teeth Movement with Time during En Masse Retraction Using Orthodontic Mini-Screw

**Jeong-Bo Hwang [1] and Sung-Seo Mo [2],***

1   Graduate School of Clinical Dental Science, The Catholic University of Korea, Seoul 06591, Republic of Korea
2   Division of Orthodontics, Department of Dentistry, Yeouido St. Mary's Hospital, College of Medicine, The Catholic University of Korea, Seoul 07345, Republic of Korea
*   Correspondence: dmoss1@hanmail.net; Tel.: +82-2-3779-1319

**Abstract:** Introduction: The aim of this study was to determine the placement of an orthodontic mini-screw (OMS) and the length of an anterior retraction hook (ARH) with en masse retraction. Continuous maxillary tooth movement pattern was simulated by finite element analysis (FEA). Materials and methods: Extraction of the first premolar was hypothesized with a finite element model. The placement of OMS was analyzed for the following two groups: (1) a high OMS (HOMS) group with OMS placed horizontally at the mesial side of the second premolar and apically 10 mm above the arch wire, and (2) a low OMS (LOMS) group with OMS placed horizontally between the second premolar and the first molar and apically 8 mm above the arch wire. According to the height of ARH, each group was divided into three subgroups. Results: When the extraction space of the first premolar was closed, anterior teeth were intruded in the HOMS group but extruded in the LOMS group. In all cases, the first molar was intruded. According to the intrusion of the first molar and extrusion of anterior teeth, the occlusal plane rotated clockwise (CW) in the LOMS group. However, in the HOMS1 group, the occlusal plane rotated counterclockwise (CCW) due to more intrusion of anterior teeth than that of the first molar. Conclusion: By analyzing six cases of different OMS and ARH, changes of incisor and molar in en masse retraction with the extraction of the first premolar could be predicted. In addition, OMS placement and ARH length can be determined based on results of incisal showing. This study can also help esthetic orthodontic results.

**Keywords:** finite element analysis; continuous teeth movement; orthodontic mini-screw; anterior retraction hook; rotation of occlusal plane; en masse retraction

## 1. Introduction

In orthodontics, en masse retraction, in which the first premolars are extracted, and an orthodontic mini-screw (OMS) is used as an anchor to pull all the anterior teeth posteriorly at once to close the extraction space, is a very common treatment for those with forward protrusion or crowding. However, studies on tooth movement patterns according to the placement position of the OMS and the height of the anterior retraction hook (ARH) are limited [1–4].

In recent studies, tooth movement for multiple teeth connected with orthodontic wire was analyzed as a finite element, with most studies limited to the initial response [1–3]. In particular, finite element studies of continuous archwire make it difficult to accurately assess tooth movement because the initial response includes not only tooth movement but also displacement due to elastic deformation of the archwire, teeth, and alveolar bone. In addition, it does not include the changes in the force system that occur as the extraction space closes, making it difficult to accurately assess teeth movement.

To minimize such problems, Kojima and Fukui [5] and Chae et al. [6] have analyzed continuous tooth movement of maxillary teeth in a non-extraction model using finite element analysis (FEA). However, there are still few studies on continuous tooth movement

along a continuous arch wire in the extraction model [7–9]. As an alternative to this, in a study of maxillary dentition by Song et al. [10], a model before the extraction space was closed (M1) and a model with the extraction space almost closed (M2) were produced to perform a finite element study on initial tooth movement at each stage. Their results showed that the pattern of tooth movement at the initial stage of extraction was different from the pattern of movement when the extraction space was closed. Therefore, a treatment plan based on the initial condition may not be appropriate.

In recent FEA, a continuous FEA method has been actively applied to overcome these limitations [5–9]. In the present study, effects of OMS placement position and ARH length on tooth movement during en masse retraction in a model in which the maxillary first premolar was extracted were analyzed through continuous FEA to help us determine suitable OMS placement position and ARH length in clinical practice.

## 2. Materials and Methods

### 2.1. Finite Element Modeling

For the finite element tooth model, a normal occlusion adult tooth model (Model-i21D-400G, Nissin Dental Products®, Kyoto, Japan) was used. Maxillary right teeth were three-dimensionally laser-scanned and created as a computer file. The arch form was modeled based on the Broad arch form of Ormco (Orange, CA, USA). The size of the bracket was modeled based on the Micro-arch (Tomy Co.®, Tokyo, Japan) stainless steel bracket [11]. A reproduction of the entire maxillary arch was made through a mirror image of the teeth on the right side of the maxilla. Finite element analysis calculated only one side and made it mirror-symmetric when imaging the result, making it possible to reduce the computation time. To make the extracted model, the maxillary first premolar was removed to complete the modeling of the tooth part. Each tooth is independent, having a contact point with an adjacent tooth through a contact point on the mesio-distal side [12]. Tooth inclination and angulation were arranged with reference to the angle as described previously. Brackets were positioned at the facial axis points of each crown [13–15]. The curve of Spee and the curve of Wilson were not established (Figure 1).

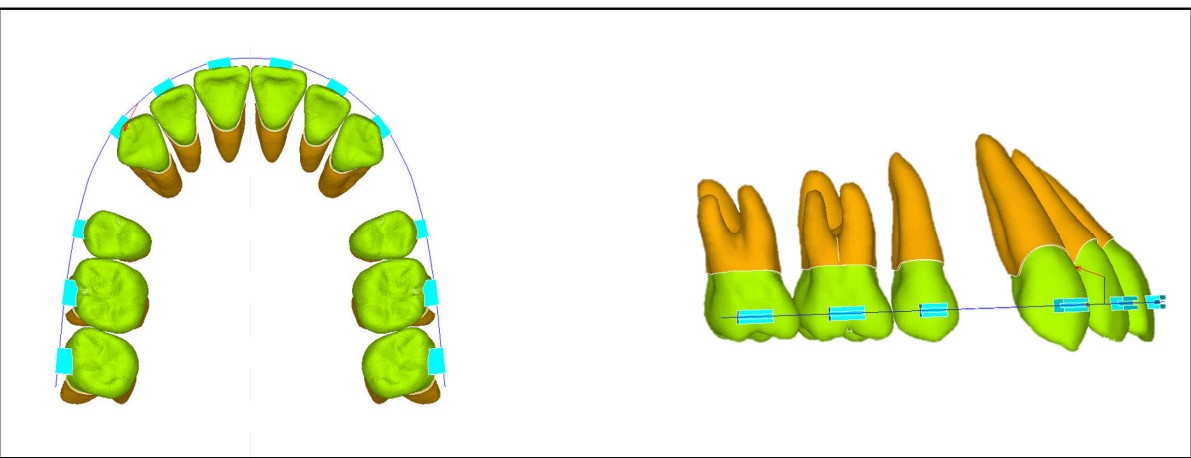

**Figure 1.** Finite element modeling of teeth, brackets, and archwire.

A stainless-steel wire (0.019 × 0.025 in) was used for the analysis. It was set to slide without friction between the arch wire and the bracket. No play was allowed between the arch wire and the bracket. The thickness of the periodontal ligament was modeled as 0.2 mm based on a study by Coolidge and Kronfeld et al. [16,17]. It was assumed to be an isotropic, homogeneous, linear elastomer. The alveolar bone was modeled so that the level was located 1 mm apical from the cementoenamel junction (CEJ) [18]. The ARH was made of stainless steel with a diameter of 0.9 mm. The retraction force was 300 g on one side. Young's modulus and Poisson's ratio (Table 1) were based on previous studies [19–22].

**Table 1.** Common material properties.

|                      | Young's Modulus (MPa) | Poisson's Ratio |
|----------------------|-----------------------|-----------------|
| Teeth                | $2.0 \times 10^4$     | 0.30            |
| Periodontal ligament | $5.0 \times 10^{-2}$  | 0.30            |
| Stainless steel      | $2.0 \times 10^5$     | 0.30            |

### 2.2. Finite Element Analysis and Simulation Conditions

Finite element analysis was performed using ANSYS 11 (Ansys Inc., Canonsburg, PA, USA). In order to realize continuous tooth movement, output data, which were calculation results of the FEA, were read again, and repeated calculations were performed by inserting previous output data into the input data. A total of six conditions were assumed by changing the OMS position and ARH length, as shown in Table 2.

**Table 2.** Location of OMS and ARH for each case.

|      |   | ARH Location | ARH Length | OMS Location | OMS Height |
|------|---|--------------|------------|--------------|------------|
|      | 1 | #12, #13     | −1 mm      | #14, #15     | +10 mm     |
| HOMS | 2 | #12, #13     | +1 mm      | #14, #15     | +10 mm     |
|      | 3 | #12, #13     | +3 mm      | #14, #15     | +10 mm     |
|      | 1 | #12, #13     | +1 mm      | #15, #16     | +8 mm      |
| LOMS | 2 | #12, #13     | +3 mm      | #15, #16     | +8 mm      |
|      | 3 | #12, #13     | +6 mm      | #15, #16     | +8 mm      |

ARH Location: Located between the teeth; ARH Length: Upward +, Downward −; OMS Location: Located between the teeth; OMS Height: Upwards from the arch wire.

The horizontal position of ARH was determined between the lateral incisor and canine commonly used clinically. The vertical length of ARH was generally located in the +Z axis direction, which was the gingival direction in the case of the maxilla. The length was −1 and +1, +3, +6 mm.

High OMS (HOMS) was determined when the OMS was positioned 10 mm upward in the +Z-axis direction from the arch wire. It was horizontally positioned in the Mesial region of the second premolar. The case of positioning at 8 mm in the +Z-axis direction from the arch wire was defined as a low OMS (LOMS). It was horizontally positioned between the second premolar and the first molar (Figure 2).

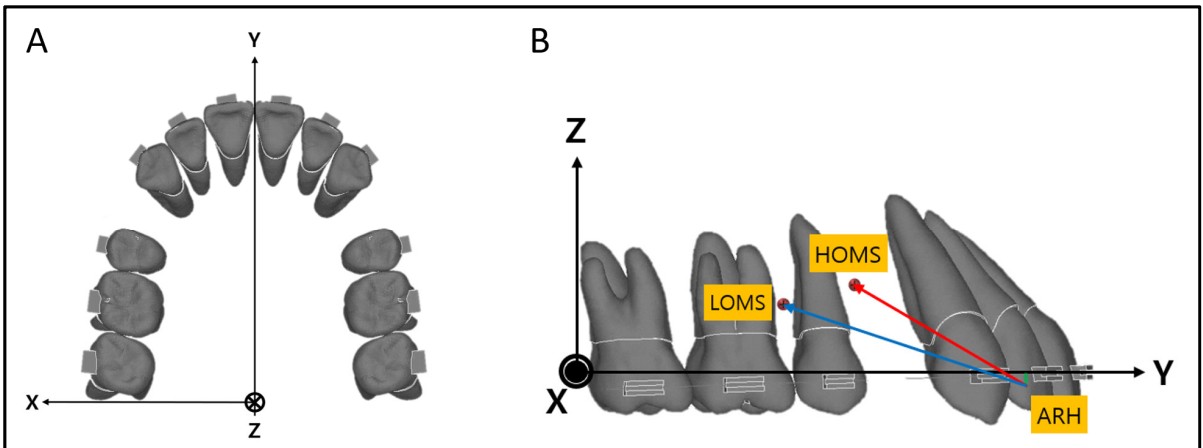

**Figure 2.** (**A**) Occlusal view of the maxillary dentition; (**B**) Lateral view of the maxillary dentition. ARH, Anterior retraction hook; HOMS, High OMS; LOMS, Low OMS.

### 2.3. Interpretation Method

In clinical practice, when teeth are subjected to orthodontic forces, displacement occurs. In this process, the periodontal ligament (PDL) was affected. Compressed and elongated

areas were created. If the orthodontic force was continued, remodeling of the alveolar bone was performed. A remodeling of the alveolar bone and balance of forces were achieved [23].

At this time, the previous position and the changed position of the tooth could be expressed by three X, Y, and Z-axis displacements and rotation about each axis. Amounts of movement and rotation are different for each point of the tooth. The center of the bracket is a part tied to the arch wire. It can be seen as a point representing each tooth. Displacement and rotation values of the center of the bracket were regarded as representative values of the corresponding teeth. They were applied for the calculation of coordinate values of the teeth after movement. Three-dimensional coordinate values before tooth movement were set as CX, CY, and CZ. Calculated values were derived as DX, DY, and DZ for displacement caused by the force of these coordinates. The amount of change of the moment, which was the amount of rotation, was RX, RY, and RZ so that the result could be displayed. Therefore, it was designed so that subsequent coordinate values could be calculated by adding six variations to previous coordinate values. This process was continuously performed.

In the anterior part, the arch wire and the bracket were attached through the point. Only rotation was allowed. All teeth were placed in contact with each other through a contact point. Conditions were given to prevent interference with each other.

The arch wire was mathematically combined with the bracket while maintaining the stiffness. A square arch wire was put in the bracket slot, with conditions similar to those of a round arch wire. The torsion of the bracket and the torsion of the arch wire were prevented from interfering with each other. Therefore, it was possible to push and pull the bracket when the arch wire was moved by force. The bracket integrated with the teeth was free to rotate under the influence of the arch wire. As a result, when force was applied to the teeth, angulation and inclination were applied similarly to those in the clinical process.

Before the orthodontic force was applied, the first arch wire was placed in the slot of the bracket without resistance. When retraction force was applied, the arch wire was bent, and the bent state was re-stretched so that it was positioned within the bracket slot of each tooth. At this time, the stretched wire was placed in the bracket slot again based on the canine teeth. The remaining teeth were also mathematically constrained so that the arch wire was positioned again in the bracket slot. At this time, each tooth was subjected to a force. As a result, each point of the finite element was changed by displacement and rotation amount. If this change was calculated and added to the previous coordinate value, a subsequent coordinate value could be obtained. This process was designed to be calculated repeatedly. The finite element analysis was repeated until the extraction space was closed. The number of iterations (IN) is shown in Table 3.

**Table 3.** Iteration number for each case.

|  |  | Iteration Number, IN |
|---|---|---|
|  | 1 | 277 |
| HOMS | 2 | 257 |
|  | 3 | 243 |
|  | 1 | 207 |
| LOMS | 2 | 203 |
|  | 3 | 197 |

## 3. Results

### 3.1. HOMS Condition

The appearance in the occlusal plane showed no significant difference in each condition of HOMS between before and after tooth movement (Figure 3A–C). When viewed from the side, it could be seen that the intrusion of the maxillary incisors increased from HOMS3 (purple) to HOMS1 (red) after space closing was completed (Figure 3D). The order of magnitude of the vertical component force in the +Z-axis direction among posterior retraction forces was HOMS1 > HOMS2 > HOMS3. As a result, in the case of HOMS1, the

force in the direction of teeth intrusion was relatively large. Thus, a lot of intrusions were observed in the maxillary anterior teeth in the HOMS1 condition.

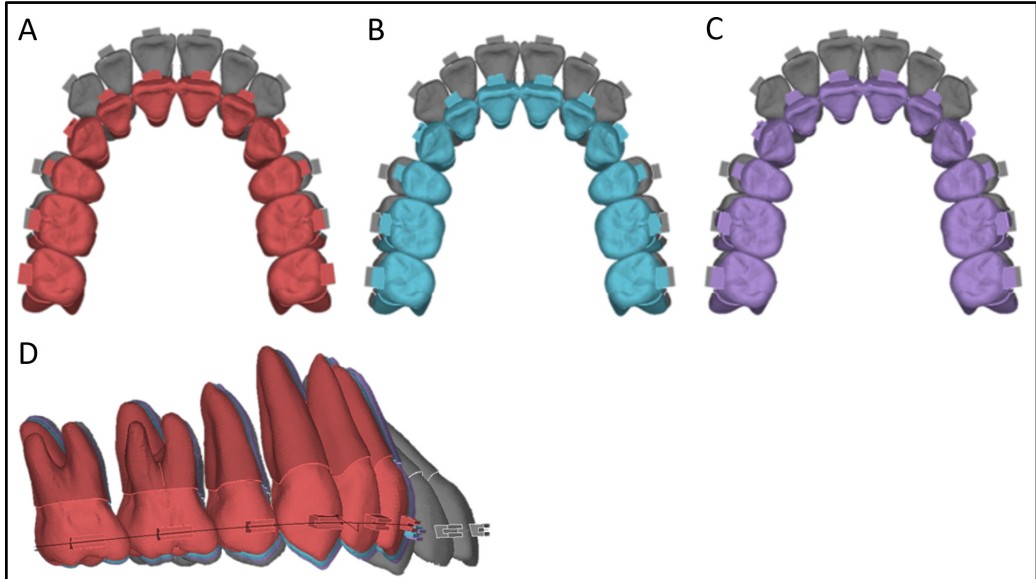

**Figure 3.** (**A**) Occlusal view of HOMS1 (Red) after space closing; (**B**) Occlusal view of HOMS2 (Sky) after space closing; (**C**) Occlusal view of HOMS3 (Violet) after space closing; (**D**) Overlap of lateral view at HOMS. Gray-colored teeth, teeth before movement.

### 3.2. LOMS Condition

The appearance of teeth in the occlusal plane did not show a significant difference in each condition of LOMS between before and after movement (Figure 4A–C). However, when viewed from the lateral side, compared with Figure 3, which was the result of HOMS, it could be seen that the anterior tooth axis was more upright, and extrusion had occurred (Figure 4D). The amount of extrusion of maxillary incisors increased from LOMS1 (orange) to LOMS3 (yellow).

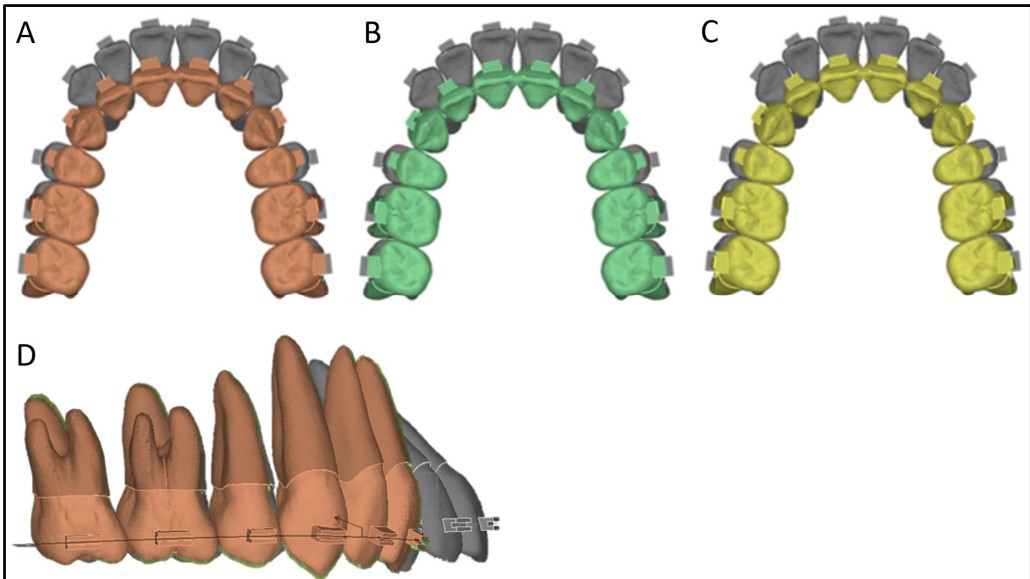

**Figure 4.** (**A**) Occlusal view of LOMS1 (Orange) after space closing; (**B**) Occlusal view of LOMS2 (Green) after space closing; (**C**) Occlusal view of LOMS3 (Yellow) after space closing; (**D**) Overlap of lateral view at LOMS. Gray-colored teeth, teeth before movement.

### 3.3. Changes in Maxillary Central Incisors

The axis of maxillary central incisors showed a tendency to be lingually tipped in all conditions (Figure 5). Amounts of changes of the root apex and incisal edge coordinate values are shown in Table 4.

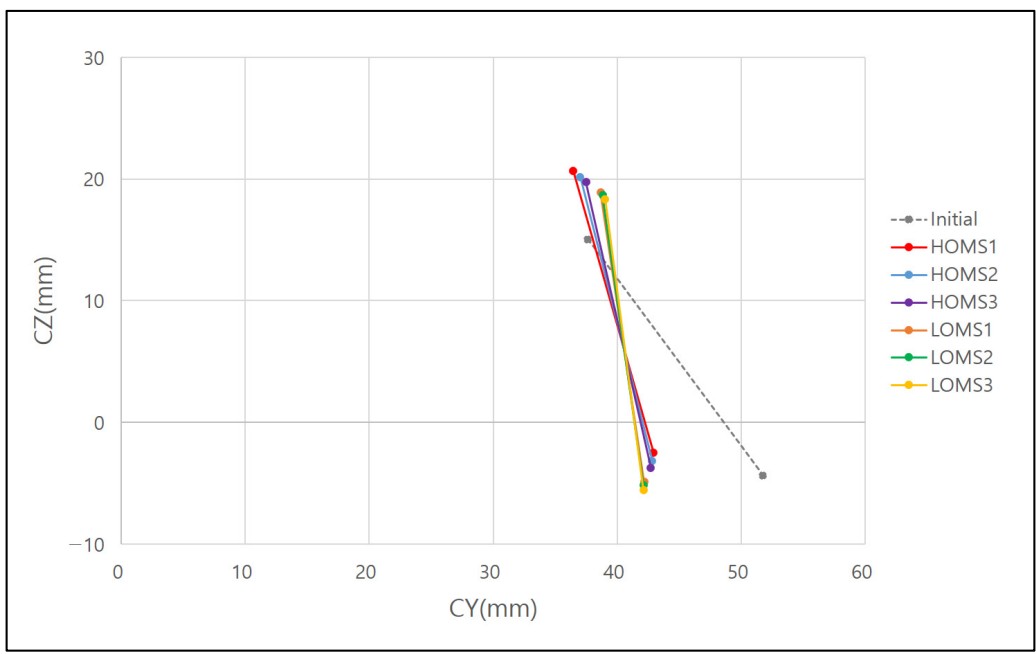

**Figure 5.** Changes in maxillary central incisor axis.

**Table 4.** Incisal edge and root apex change in maxillary central incisor for each case.

|  | HOMS1 | | HOMS2 | | HOMS3 | |
|---|---|---|---|---|---|---|
|  | ΔCY (mm) | ΔCZ (mm) | ΔCY (mm) | ΔCZ (mm) | ΔCY (mm) | ΔCZ (mm) |
| Incisal edge | −8.82 | 1.85 | −8.92 | 1.19 | −9.05 | 0.639 |
| Root apex | −1.18 | 5.59 | −0.603 | 5.11 | −0.148 | 4.70 |
|  | LOMS1 | | LOMS2 | | LOMS3 | |
|  | ΔCY (mm) | ΔCZ (mm) | ΔCY (mm) | ΔCZ (mm) | ΔCY (mm) | ΔCZ (mm) |
| Incisal edge | −9.58 | −0.530 | −9.63 | −0.803 | −9.62 | −1.19 |
| Root apex | 1.02 | 3.86 | 1.19 | 3.61 | 1.39 | 3.25 |

$\theta = tan^{-1} \frac{\Delta CZ}{\Delta CY}$ and values obtained by converting $\theta$, a Radian value, into degree (°) are summarized in Table 5. The angle before tooth movement was about 54°. However, from HOMS1 to LOMS3, it could be seen that the angle of the tooth axis gradually increased and tended to stand upright.

**Table 5.** Inclination of maxillary central incisor for each case.

|  | Incisal Inclination (Deg.) | Differences from Initial (Deg.) |
|---|---|---|
| Initial | 53.98 | - |
| HOMS1 | 74.34 | 20.36 |
| HOMS2 | 76.03 | 22.06 |
| HOMS3 | 77.46 | 23.48 |
| LOMS1 | 81.59 | 27.61 |
| LOMS2 | 82.10 | 28.12 |
| LOMS3 | 82.55 | 28.57 |

Figure 5 shows the position of the incisal edge of the central incisor before tooth movement and after space closing. In the case of HOMS, compared to the initial state, it was found that the incisal edge of the maxillary central incisor intruded. However, in the case of LOMS, it was found that the incisal edge of the maxillary central incisor extruded conversely. It could be seen that the degree of intrusion decreased from HOMS1 to HOMS3, whereas the degree of extrusion increased from LOMS1 to LOMS3.

Through the amount of change of the incisal edge CY value (ΔCY), it could be seen that the incisal edge of the maxillary central incisor had moved posteriorly about 8.82~9.63 mm (average 9.27 mm) until space closure (Table 4).

Figure 6 shows vertical position changes of the incisal edge of maxillary incisors according to IN to examine how the continuous tooth movement progressed. In the case of LOMS, the maxillary anterior teeth were slightly extruded as a result. It could be seen that the shorter the hook length, the smaller the extrusion amount. It also could be seen that the longer the length of the hook, the smaller the IN until space closure. In the case of HOMS, it was confirmed that the maxillary anterior teeth were intruded as a result. It was observed that the vertical position of the maxillary anterior incisal edge reached the lowest point on the Z-axis at the time of IN about 20 and then increased again. In the case of LOMS, it could be seen that the graphs of the three cases had an intersection point at the time of the 30th calculation and that the location of the lowest point was very different in each condition. Looking at the tendency, the case of LOMS1 had the lowest point in the front part, and the position of the lowest point moved to the posterior part as it went to LOMS3. It bounced in all conditions of LOMS. However, it was flat compared to HOMS.

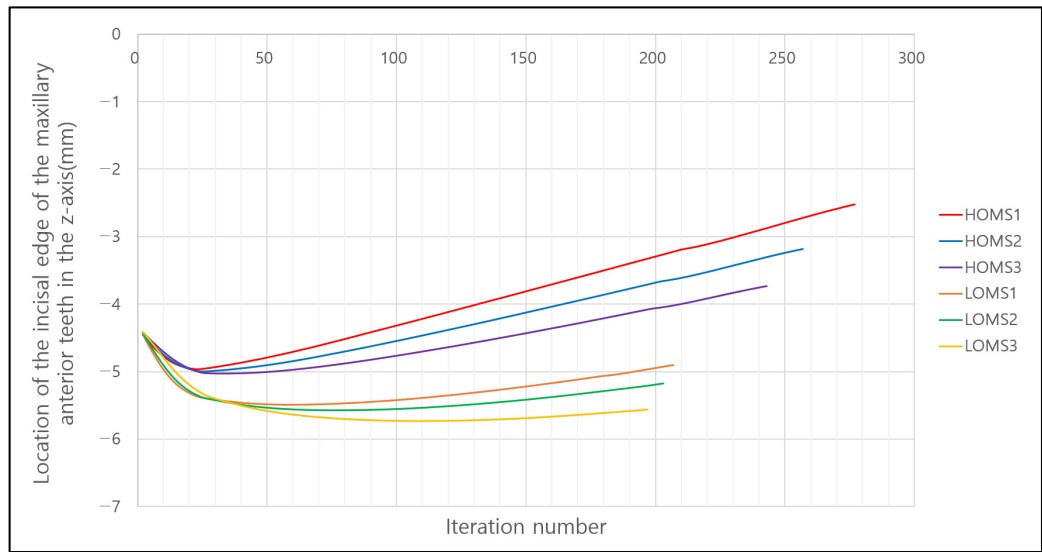

**Figure 6.** Vertical position of maxillary central incisor edge.

### 3.4. Changes in Maxillary First Molars

The amount of change in the maxillary first molars can be seen in Figure 7 and Table 6. The tooth axis was calculated using the coordinates of the mesial palatal cusp and the palatal root apex. In HOMS, it could be seen that the tooth axis was inclined in a counterclockwise (CCW) direction and that the degree was decreased from HOMS1 to HOMS3. In LOMS, it could be seen that the tooth axis was inclined in a clockwise (CW) direction and that the degree became larger as it went from LOMS1 to LOMS3. Rotation of the occlusal plane is expected to be the biggest factor that causes changes in maxillary first molars, which will be discussed later.

Whether the maxillary first molar mesio-palatal cusp tip extruded or intruded, compared to the initial position, it was intruded in all cases (Figure 7). Additionally, it can be seen that the degree of intrusion decreases from HOMS1 to LOMS3. The mesio-palatal cusp of the maxillary first molar intruded by 1.00~1.51 mm (mean 1.26 mm) (Table 6).

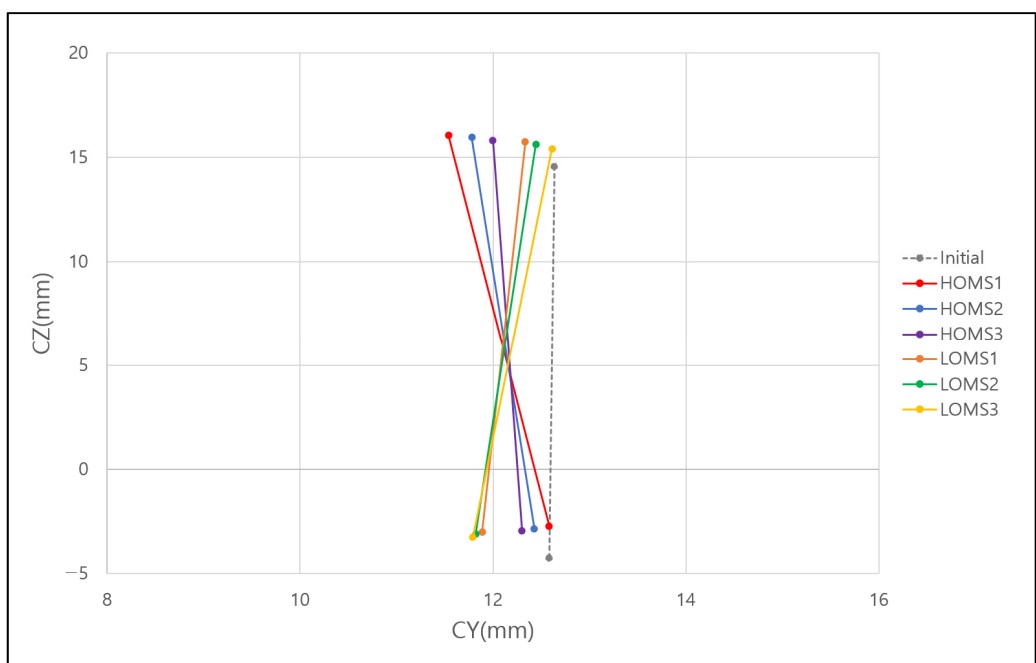

**Figure 7.** Changes in tooth axis of maxillary first molar after initial tooth movement and completion of space closure.

**Table 6.** Changes in maxillary first molar's mesio-palatal cusp tip and root apex until the extraction space is closed for each case.

| | HOMS1 | | HOMS2 | | HOMS3 | |
|---|---|---|---|---|---|---|
| | $\Delta$CY (mm) | $\Delta$CZ (mm) | $\Delta$CY (mm) | $\Delta$CZ (mm) | $\Delta$CY (mm) | $\Delta$CZ (mm) |
| Mesio palatal cusp tip | $-1.45 \times 10^{-3}$ | $1.52 \times 10^{0}$ | $-1.53 \times 10^{-1}$ | $1.39 \times 10^{0}$ | $-2.83 \times 10^{-1}$ | $1.29 \times 10^{0}$ |
| Root apex | $-1.10 \times 10^{0}$ | $1.50 \times 10^{0}$ | $-8.56 \times 10^{-1}$ | $1.39 \times 10^{0}$ | $-6.38 \times 10^{-1}$ | $1.27 \times 10^{0}$ |
| | LOMS1 | | LOMS2 | | LOMS3 | |
| | $\Delta$CY (mm) | $\Delta$CZ (mm) | $\Delta$CY (mm) | $\Delta$CZ (mm) | $\Delta$CY (mm) | $\Delta$CZ (mm) |
| Mesio palatal cusp tip | $-6.95 \times 10^{-1}$ | $1.23 \times 10^{0}$ | $-7.61 \times 10^{-1}$ | $1.14 \times 10^{0}$ | $-7.93 \times 10^{-1}$ | $1.00 \times 10^{0}$ |
| Root apex | $-3.04 \times 10^{-1}$ | $1.19 \times 10^{0}$ | $-1.93 \times 10^{-1}$ | $1.06 \times 10^{0}$ | $-2.64 \times 10^{-2}$ | $8.51 \times 10^{-1}$ |
| | HOMS1 | HOMS2 | HOMS3 | LOMS1 | LOMS2 | LOMS3 |
| Mesio palatal cusp tip $\Delta$CZ (mm) | +1.52 | +1.39 | +1.29 | +1.23 | +1.14 | +1.00 |
| Change of Mesio palatal cusp | Intrusion | Intrusion | Intrusion | Intrusion | Intrusion | Intrusion |

$\Delta$CZ (mm): + means intrusion, − means extrusion.

*3.5. Change in the Angle of the Occlusal Plane*

The angular change of the occlusal plane was investigated through the coordinate values of the incisal edge of the maxillary central incisor and the mesio-palatal cusp of the maxillary first molar. The position of the maxillary first molar mesio-palatal cusp intruded 1.00~1.51 mm in the +Z direction, and the incisal edge of the maxillary central incisor extruded or intruded by −1.19~+1.85 mm in the +Z direction under six conditions (Figure 8). The initial occlusal plane angle was −0.16°. After space closure, only HOMS1 had a positive value (CCW) of +0.42°. Under the remaining conditions, the occlusal plane angle after space closure had a negative value (CW). The absolute value increased from HOMS2 to LOMS3 (Table 7).

In the case of LOMS, according to the intrusion of the first molar and extrusion of the anterior teeth, the occlusal plane rotated clockwise (CW). However, in the HOMS1 group,

counterclockwise (CCW) rotation of the occlusal plane was observed due to more intrusion of the anterior teeth than that of the first molar.

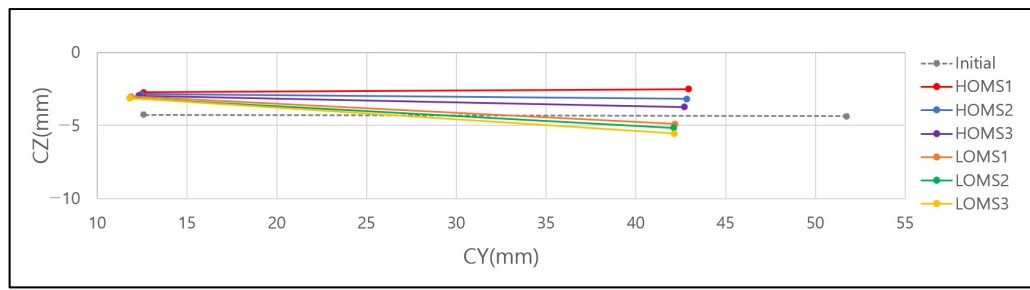

**Figure 8.** Changes of occlusal plane.

**Table 7.** Angle and difference in angle value of occlusal plane after initial and space closing.

| | Angle of Initial Occlusal Plane (Deg.) | Angle of Occlusal Plane after Space Closing (Deg.) | Angle Differences (Deg.) |
|---|---|---|---|
| HOMS1 | | +0.42 | +0.58 |
| HOMS2 | | −0.60 | −0.43 |
| HOMS3 | | −1.44 | −1.27 |
| LOMS1 | −0.16 | −3.54 | −3.37 |
| LOMS2 | | −3.87 | −3.71 |
| LOMS3 | | −4.34 | −4.17 |

+ means CCW rotation; − means CW.

## 4. Discussion

Many finite element studies have looked at changes in teeth based on their initial responses when an orthodontic force is applied [1–3]. However, during orthodontic treatment, tooth movement occurs, and the force system changes accordingly. To compensate for these shortcomings, Song et al. [10] have conducted a study by dividing the start and end time points of tooth movement. In the present study, the space closing in the maxillary first premolar extraction model was simulated through continuous finite element analysis.

As a result of the study by Song et al. [10], in the case of HOMS, the incisal edge of the maxillary central incisor extruded in M1 (model immediately after extraction model). However, in M2 (model with 1 mm of extraction space left), the incisal edge of the maxillary central incisor intruded. These results could be confirmed through a continuous process of changes in the present study. Moreover, in LOMS, extrusion of the incisal edge of the maxillary central incisors was observed in both M1 and M2 models, and the continuous process of changes was confirmed in this study (Figure 6). The axis of the maxillary central incisors showed a more upright tendency in LOMS (Figure 5), which was also consistent with the results of Song et al. [10]. The rotation of the occlusal plane in CW in the M1 model was also confirmed through the tooth movement pattern before 'bounce,' as shown in Figure 6. In LOMS, the occlusal plane was rotated in CW larger than in HOMS. In the M2 model, the rotation of the occlusal plane by CCW for HOMS and CW for LOMS was also consistent with the results of the present study. If there was a difference between the results of Song et al. [10] and the results of the present study, rotation of the occlusal plane was observed with CCW only in HOMS1 in the present study. The reason for this difference might be because a significant amount of first molar intrusion was observed in continuous finite element analysis. In the M2 condition, the maxillary central incisors intruded in all HOMS conditions. In the case of HOMS1, which had the largest vertical component of retraction force due to the shortest ARH (−1 mm), CCW rotation of the occlusal plane was found, as shown in Figure 8.

In a study by Lim [24], HOMS was used for patients with gummy smiles, ARH was placed at −1 mm, and en masse retraction was performed to complete the treatment

aesthetically. In the FEA results of the present study, intrusion of the maxillary central incisor was the largest in the ARH −1 mm condition of HOMS1, and the occlusal plane was rotated the most by CCW. Therefore, actual clinical results and FEA analysis results were consistent.

Lee et al. [25] have studied the effect of OMS position on tooth movement. When the OMS was placed mesial to the second premolar and a short hook (+1 mm) was used, the maxillary central incisor was moved posteriorly by 7.23 mm and intruded by 1.59 mm after space closing, similar to results of our study (HOMS2: the maxillary central incisor was moved posteriorly by 8.92 mm and intruded by 1.19 mm after space closing). When the OMS was placed between the first molar and the second premolar, the maxillary central incisor was moved posteriorly by 7.2 mm and extruded by 0.25 mm after space closing, similar to results of our study (LOMS1: the maxillary central incisor was moved posteriorly by 9.58 mm and extruded by 0.53 mm after space closing).

$\vec{M}$ (moment) has a $d \times \vec{F}$ relationship, where d is the shortest distance from CR (center of resistance). On our coordinate axis, the center of resistance (anterior teeth's CR, here in after aCR) of the 6 maxillary incisors, is located at 14.0 mm in the −Y-axis direction and 13.5 mm in the +Z-axis direction around the incisal edge of the maxillary central incisors [12]. The center of resistance (posterior teeth's CR, here in after pCR) of the posterior molars, according to the study of Kojima et al. [7], is shown in Figure 9. Moments created by anterior and posterior teeth were generated around these points. The sum of these moments became the moment of the entire tooth as a result. Calculating the moment based on tooth position in the initial state is shown in Table 8. In the case of HOMS, since total moments has a positive value, it tends to rotate in a CCW direction. In the case of LOMS, it tends to rotate in a CW direction because it has a negative value.

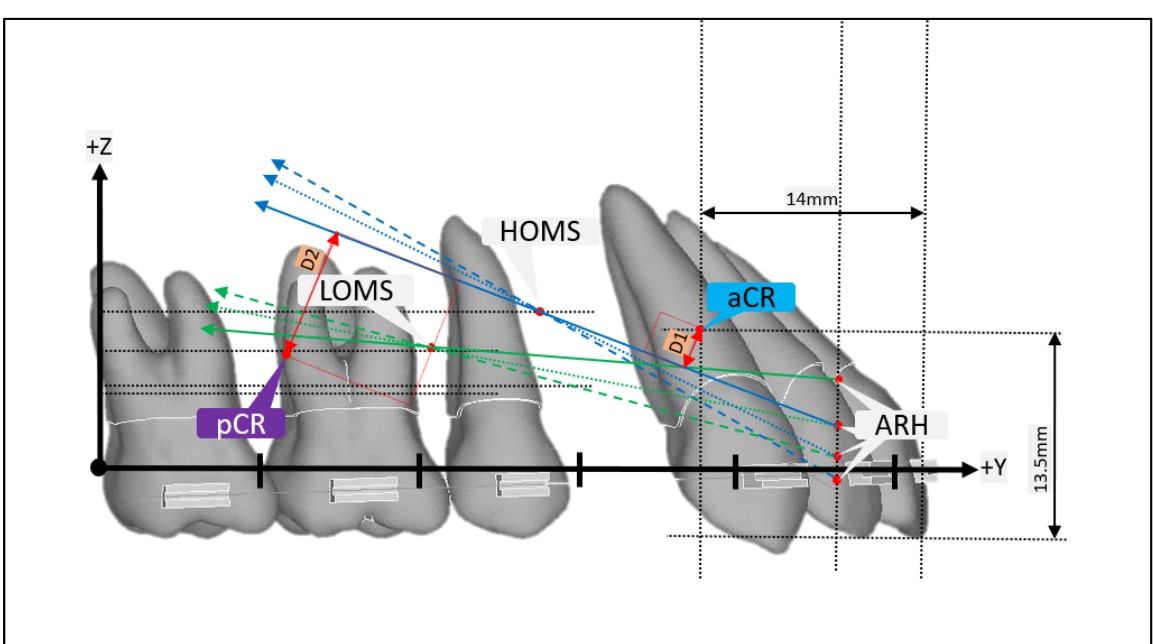

**Figure 9.** A schematic diagram of the CR and the Force vector in Initial. aCR, anterior teeth's CR; pCR, posterior teeth's CR; D1, the shortest distance between the traction force vector and aCR; D2, the shortest distance between traction force vector and pCR.

The rotation of the occlusal plane appears to be a combination of the sum of moments generated by anterior and posterior teeth and intrusion or extrusion of the teeth by the vertical component of the traction force.

As a result, in this study, the occlusal plane was rotated in the CCW direction only in HOMS1. For patients with a gummy smile, the HOMS1 condition is expected to show the best effect. In the case of patients with vertical overgrowth in the face, maxillary

molars often need to be intruded. This is also expected to show the best effect in the HOMS1 condition. However, due to the intrusion of maxillary molars, the mandible will rotate CCW, and the tip of the chin will protrude forward-upward [26], which may lead to undesirable treatment results for Class III patients. To perform en masse retraction while maintaining the axis of maxillary incisors, it seems necessary to have retraction by giving a compensating curve to the arch wire [22].

**Table 8.** Magnitude and direction of moment at initial time.

| | D1 (mm) | Force (gram·f) | M1 (gram·f·mm) | Rotation (CW, −; CCW, +) |
|---|---|---|---|---|
| HOMS1 | 3.93 | 300 | 1180.09 | − |
| HOMS2 | 3.38 | 300 | 1014.88 | − |
| HOMS3 | 2.64 | 300 | 790.66 | − |
| LOMS1 | 5.47 | 300 | 1640.33 | − |
| LOMS2 | 4.37 | 300 | 1309.91 | − |
| LOMS3 | 2.36 | 300 | 708.06 | − |
| | **D2 (mm)** | **Force (gram·f)** | **M2 (gram·f·mm)** | **Rotation (CW, −; CCW, +)** |
| HOMS1 | 10.03 | 300 | 3009.24 | + |
| HOMS2 | 9.20 | 300 | 2761.42 | + |
| HOMS3 | 8.30 | 300 | 2490.00 | + |
| LOMS1 | 2.75 | 300 | 826.07 | + |
| LOMS2 | 2.05 | 300 | 613.65 | + |
| LOMS3 | 1.10 | 300 | 330.43 | + |
| | **M1 (gram·f·mm)** | **M2 (gram·f·mm)** | **Total M (gram·f·mm)** | |
| HOMS1 | −1180.09 | +3009.24 | +1829.15 | |
| HOMS2 | −1014.88 | +2761.42 | +1746.54 | |
| HOMS3 | −790.66 | +2490.00 | +1699.34 | |
| LOMS1 | −1640.33 | +826.07 | −814.27 | |
| LOMS2 | −1309.91 | +613.65 | −696.26 | |
| LOMS3 | −708.06 | +330.43 | −377.63 | |

In patients with insufficient incisal showing, the opposite treatment technique should be chosen. In this case, strategies for extruding maxillary incisors should be chosen, and the most suitable condition is considered to be LOMS3. It is expected that the intrusion of the posterior region will occur, and a posterior settling process may be required at the end of treatment. In addition, the possibility of traumatic occlusion of maxillary anterior and mandibular incisors due to rotation of the occlusal plane to CW should be considered.

The last thing to look at is the lingual tipping of the posterior and anterior regions. According to FEA results, the palatal cusp of posterior teeth after space closure was not clearly visible. The palatal cusp of posterior teeth moved more in the +Z-axis direction than buccal cusps. This was because when the arch wire received retraction force and moved backward while closing the extraction space, a force to narrow the width of the posterior teeth occurred, and the bowing phenomenon of the arch as anterior teeth were inclined to the lingual side by the retraction force. In the present study, central incisors were rotated at the lingual side by 20.36 to 28.57 degrees. In actual clinical practice, there is a tendency for the anterior tooth to be inclined to the lingual side during en masse retraction, which seems to have been well implemented in this FEA (Figure 10).

Although FEA is a research method applied in a planned model with an ideal arrangement, it has the limitation of not reflecting individualized patient conditions such as tooth arrangement, alveolar bone condition, masticatory pressure, or chewing habit. Through continuous FEA in our study, it was confirmed that many parts of tooth movement that could be observed in actual clinical practice were implemented in FEA. Further clinical studies are needed to support this finding.

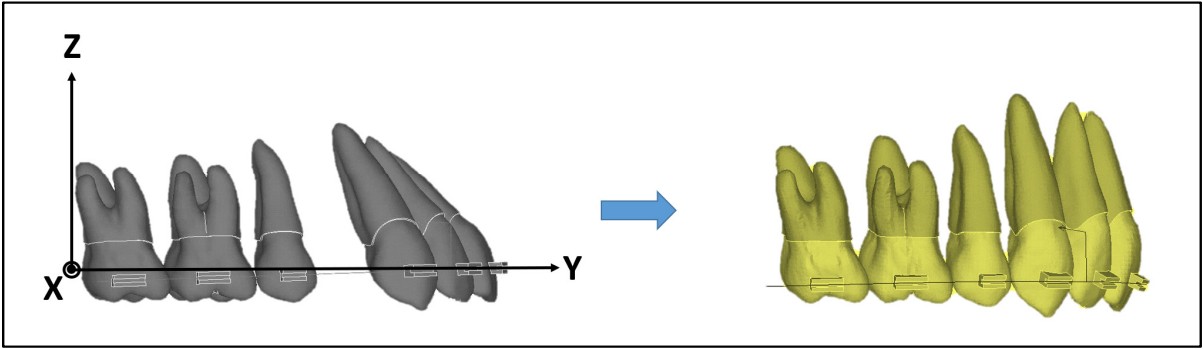

**Figure 10.** Tooth movement of posterior and anterior teeth after space closing in LOMS3.

**5. Conclusions**

In our continuous FEA study using the first premolar extraction case as a reference model, it was confirmed that movement patterns of dentition were different depending on the length of ARH and the position of OMS during en masse retraction. In the HOMS group, intrusion of the incisal edge of the maxillary anterior teeth was observed. In the LOMS group, extrusion of the incisal edge of the maxillary anterior teeth was observed. With shorter ARH, more intrusion of the maxillary anterior teeth was observed.

**Author Contributions:** Conceptualization, J.-B.H. and S.-S.M.; methodology, J.-B.H. and S.-S.M.; software, J.-B.H.; validation, J.-B.H. and S.-S.M.; formal analysis, J.-B.H. and S.-S.M.; writing—original draft preparation, J.-B.H.; writing—review and editing, J.-B.H. and S.-S.M.; visualization, J.-B.H.; supervision, S.-S.M.; project administration, S.-S.M.; All authors have read and agreed to the published version of the manuscript.

**Funding:** This research received no external funding.

**Institutional Review Board Statement:** Not applicable.

**Informed Consent Statement:** Not applicable.

**Data Availability Statement:** The authors declare that the materials are available.

**Conflicts of Interest:** The authors declare no conflict of interest.

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
