# Peer review of "Finite Element Analysis of Maxillary Teeth Movement with Time during En Masse Retraction Using Orthodontic Mini-Screw"

_applsci, doi:10.3390/app13063579_

Round 1

Reviewer 1 Report

In my oppinion i would suggest the authors to move table 8 and figure 9 in the results section.

Please present also the limitations of this study.

In the discussion i suggest the authors to compare their results with the result from other similar reasearch articles more recent published in the last 5 years.

Author Response

Thank you for your kind comments.

In my opinion i would suggest the authors to move table 8 and figure 9 in the results section

è Table 8 and figure 9 are not the results of this study. The results of this study were interpreted based on the results obtained by drawing the force system based on previously known data (CR position). Therefore, it is considered appropriate to include it in the discussion section.

Please present also the limitations of this study.

è I will add the limitations of this study to the discussion.

In the discussion i suggest the authors to compare their results with the result from other similar reasearch articles more recent published in the last 5 years.

è The finite element study used in this study uses continuous FEA (called long term simulation in some literature) to overcome the limitations of existing finite element studies, so there are not many studies. We've added a paper to the discussion that compares our findings to clinical studies. And Recent literature using a similar method will be added as a reference.

Reviewer 2 Report

Thank you for the opportunity to pre-publish the manuscript: "Finite element analysis of maxillary teeth movement with time during En-masse retraction using orthodontic mini-screw". I hope that my remarks will be helpful for the authors.

Introduction

- The journal Applied Sciences has a large audience, many of whom are not orthodontists, but may still be interested in this application of finite element analysis. Therefore, the first paragraph should be expanded to explain why the first premolar is extracted and what En-masse retraction is.

- There is no clearly defined subsection of the purpose of the study.

Materials and methods

- Failure to include the Spee curve is, in my opinion, a far-reaching simplification. Nevertheless, I guess that the authors made such a decision, because it would be difficult to propose an averaged shape of this curve. I also suspect that the results of the study could differ significantly in the case of even a slight change in the original position of the teeth. I believe that the Authors are best able to explain these issues and should do so at the end of the Discussions section, in the Limitations subsection.

Results

- Graphs, especially in Figures 5, 7 and 8, are difficult to read due to overlapping lines. I suggest you include high resolution images so that they are available for readers to download, e.g. as Supplementary Materials.

Discussion

- It is necessary to supplement this section with references to the latest research.

Conclusions

- The conclusions are to be the answer to the research question posed. The current content of the conclusions does not justify the purpose of the study and does not seem worthy of being made public. I am asking the Authors to formulate specific clinical implications in this section, e.g. placing the mini-implant in such-and-such position leads to such-and-such tooth movement.

Author Response

Thank you for your kind comments.

Introduction

- The journal Applied Sciences has a large audience, many of whom are not orthodontists, but may still be interested in this application of finite element analysis. Therefore, the first paragraph should be expanded to explain why the first premolar is extracted and what En-masse retraction is.

è The reason for tooth extraction and en-masse retraction will be described in the first paragraph.

- There is no clearly defined subsection of the purpose of the study.

==>At the end of the introduction, the purpose of this study is described because the purpose of this study is to help clinical application by analyzing the effect of the OMS placement location and ARH length on en-masse retraction using continuous FEA.

Materials and methods

- Failure to include the Spee curve is, in my opinion, a far-reaching simplification. Nevertheless, I guess that the authors made such a decision, because it would be difficult to propose an averaged shape of this curve. I also suspect that the results of the study could differ significantly in the case of even a slight change in the original position of the teeth. I believe that the Authors are best able to explain these issues and should do so at the end of the Discussions section, in the Limitations subsection.

==> The materials in this study did not originate from the patient, and the shape of individual teeth was obtained from a dental model (Dentiform available on the market) and the individual teeth were arranged according to the normal arch form.

 Therefore no curve of speed was granted. These data are studies using average data and have limitations in not reflecting individual patient differences. This part will be described as the limitatation of the study in the discussion section.

==>We have added the limitations of this study to the discussion.

Results

- Graphs, especially in Figures 5, 7 and 8, are difficult to read due to overlapping lines. I suggest you include high resolution images so that they are available for readers to download, e.g. as Supplementary Materials.

==> Increased the resolution of the picture, but the problem of overlapping lines seems difficult to solve.

Discussion

- It is necessary to supplement this section with references to the latest research.

==>The finite element study used in this study uses continuous FEA (called long term simulation in some literature) to overcome the limitations of existing finite element studies, so there are not many studies. We've added a paper to the discussion that compares our findings to clinical studies. And Recent literature using a similar method will be added as a reference.

Conclusions

- The conclusions are to be the answer to the research question posed. The current content of the conclusions does not justify the purpose of the study and does not seem worthy of being made public. I am asking the Authors to formulate specific clinical implications in this section, e.g. placing the mini-implant in such-and-such position leads to such-and-such tooth movement.

==> Corrected the conclusion.

Reviewer 3 Report

Manuscript ID: applsci-2240449

Reviewer’s comments:

Point 1.

As far as sagittal direction is concerned, why is the HOMS nail implant located at the proximal side of the second premolar, but the LOMS implant located at the distal side of the second premolar?

Point 2.

The author cited: "The rotation of the occlusal plane seems to be determined by the combined influence of three factors." 1) The sum of the moments generated around the CR of each of the anterior 303 and posterior regions; 2) The intrusion of the teeth by the vertical component of the retraction force; 3) The lingual inclination of the maxillary incisors; however, the relationship between these three elements is not well clarified in the article.

Author Response

Thank you for your kind comments.

Point 1.

As far as sagittal direction is concerned, why is the HOMS nail implant located at the proximal side of the second premolar, but the LOMS implant located at the distal side of the second premolar?

==> The purpose of this study is to provide clinical guidance by studying the effect of the placement location of OMS and the length of ARH on en-masse retraction through finite element analysis.

In an actual clinical study (Am J orthod DO 2011;140:224-32), completely different clinical results have been reported by OMS placed in the posterior and anterior 2nd premolars.

Point 2.

The author cited: "The rotation of the occlusal plane seems to be determined by the combined influence of three factors." 1) The sum of the moments generated around the CR of each of the anterior and posterior regions; 2) The intrusion of the teeth by the vertical component of the retraction force; 3) The lingual inclination of the maxillary incisors; however, the relationship between these three elements is not well clarified in the article.

==> Edited for clarity for readers.

  • The sum of the moments generated around the CR of each of the anterior and posterior regions;==> figure 9 and table8
  • The intrusion of the teeth by the vertical component of the retraction force; ==> table 4, table 6, figure 5, figure 6, figure 7, figure 8

Reviewer 4 Report

The article "Finite element analysis of maxillary teeth movement with time during En-masse retraction using orthodontic mini-screw" aims to determine placement of orthodontic mini-screw and length of anterior retraction hook with en-masse retraction, by means of finite element analysis.

General comments:

In my opinion the introduction should be lenghtened, and the subject should be better put into perspective.

The discussion part is based on only three literature studies: Song et al. (also described in the introduction), Lim, Kojima et al.(also described in the introduction), which is not enough. I don't see the need of Figure 9 and Table 8, which show the results of the Kohima study. 

Point-by-point comments:

It's En-masse or en-masse, please choose the proper one.

Please remove the headings from the abstract.

Use Figure instead of Fig.

Please use MDPI style for the tables, for the figures caption and tables title.

Is Figure 9 original or it belongs to the study of Kojima et al.?

Line 327. By 'In this FEA study" do you mean your study? Please be more specific.

Please prepare your references list according to MDPI style.

Author Response

Thank you for your kind comments.

General comments:

In my opinion the introduction should be lenghtened, and the subject should be better put into perspective.

==>Added content to the introduction.

The discussion part is based on only three literature studies: Song et al. (also described in the introduction), Lim, Kojima et al.(also described in the introduction), which is not enough. I don't see the need of Figure 9 and Table 8, which show the results of the Kohima study.

==>We've added a paper to the discussion that compares our findings to clinical studies. (Lee’s study)

Point-by-point comments:

It's En-masse or en-masse, please choose the proper one.

==> We've unified it to en-masse

Please remove the headings from the abstract.

==> Corrected.

Use Figure instead of Fig.

==> Corrected.

Please use MDPI style for the tables, for the figures caption and tables title.

==> I edited it to fit the format.

Is Figure 9 original or it belongs to the study of Kojima et al.?

==> Kojima's study only referenced the location of the center of resistance of the posterior molars.

We analyzed the force system in this research model by referring to the location of the center of resistance of the posterior molars in Kojima's research and the center of resistance of the anterior teeth in Jeong's research.

Line 327. By 'In this FEA study" do you mean your study? Please be more specific.

==> “In present study”

Please prepare your references list according to MDPI style.

==> I edited it to fit the format.

Round 2

Reviewer 2 Report

The manuscript has been revised in line with my recommendations.

Author Response

Thank you.
Some English corrections have been made. Edited sections are colored red.
Revised the references to ACS reference style as requested by another reviewer.

Reviewer 4 Report

Once again, please use the MDPI style for the references list.

Author Response

Thank you.
Some English corrections have been made. Edited sections are colored red.
Revised the references to ACS reference style.